# In Silico Identification of Multi-Target Ligands as Promising Hit Compounds for Neurodegenerative Diseases Drug Development

**DOI:** 10.3390/ijms232113650

**Published:** 2022-11-07

**Authors:** Petko Alov, Hristo Stoimenov, Iglika Lessigiarska, Tania Pencheva, Nikolay T. Tzvetkov, Ilza Pajeva, Ivanka Tsakovska

**Affiliations:** 1Institute of Biophysics and Biomedical Engineering, Bulgarian Academy of Sciences, Acad. G. Bonchev Str., Bl. 105, 1113 Sofia, Bulgaria; 2Institute of Molecular Biology “Acad. Roumen Tsanev”, Bulgarian Academy of Sciences, Acad. G. Bonchev Str., Bl. 21, 1113 Sofia, Bulgaria

**Keywords:** neurodegenerative diseases, acetylcholinesterase, histone deacetylase 2, monoamine oxidase B, multi-target-directed ligands, virtual screening, docking, pharmacophore, molecular dynamics

## Abstract

The conventional treatment of neurodegenerative diseases (NDDs) is based on the “one molecule—one target” paradigm. To combat the multifactorial nature of NDDs, the focus is now shifted toward the development of small-molecule-based compounds that can modulate more than one protein target, known as “multi-target-directed ligands” (MTDLs), while having low affinity for proteins that are irrelevant for the therapy. The in silico approaches have demonstrated a potential to be a suitable tool for the identification of MTDLs as promising drug candidates with reduction in cost and time for research and development. In this study more than 650,000 compounds were screened by a series of in silico approaches to identify drug-like compounds with predicted activity simultaneously towards three important proteins in the NDDs symptomatic treatment: acetylcholinesterase (AChE), histone deacetylase 2 (HDAC2), and monoamine oxidase B (MAO-B). The compounds with affinities below 5.0 µM for all studied targets were additionally filtered to remove known non-specifically binding or unstable compounds. The selected four hits underwent subsequent refinement through in silico blood-brain barrier penetration estimation, safety evaluation, and molecular dynamics simulations resulting in two hit compounds that constitute a rational basis for further development of multi-target active compounds against NDDs.

## 1. Introduction

Neurodegenerative diseases (NDDs) are age-related and share common underlying characteristics such as chronic, irreversible and progressive neuronal degradation in specific brain regions. Many complex pathophysiological processes are involved in NDD progression, including oxidative stress, neuroinflammation, misfolding and aggregation of insoluble deposits (proteins) in the brain, mitochondrial dysfunction, proteolytic stress, and others [1,2,3]. Alzheimer’s disease (AD) and Parkinson’s disease (PD) are the most prevalent among the NDDs, affecting more than 8% of adults aged ≥65 years worldwide [4,5].

The common treatment of NDDs such as AD, PD, and related dementias, is mainly based on usage of single-target-directed small molecules that can be used either as mono or as combined therapy at early or late stages of the NDDs. In particular, the majority of the therapeutic approaches for the treatment of PD are focused on increasing the dopamine (DA) levels in the brain of affected patients [6]. For example, the “gold standard” levodopa (L-DOPA) replacement therapy in combination with monoamine oxidase B (MAO-B) and/or catechol-*O*-methyltransferase (COMT) inhibitors, DA agonists, or DOPA-decarboxylase inhibitors still remains the most widely used treatment of PD [6,7,8]. However, the long-term treatment with l-DOPA is associated with several side effects such as dyskinesia, on-off effects, etc. [9]. Other examples include small-molecule-based drugs for the treatment of AD, such as donepezil, galantamine, rivastigmine, and tacrine, mainly acting as acetylcholinesterase (AChE) inhibitors [5].

The above mentioned single-target-directed drugs selectively act on biological targets related to one or several symptoms at different disease stages and do not change the progressive course of the neurodegeneration in the respective brain regions of affected patients [6,7]. Therefore, there is a medical need of new therapeutic approaches with considerable disease-modifying features. Recent research shows involvement of histone-deacetylase family of enzymes (HDAC) in the regulation of memory processes in the mammal brain [10,11,12] mainly via modulation of the gene expression by histone acetylation [13]. Thus, the inhibitors of HDACs, and particularly HDAC2, are considered as potential disease-modifying agents in NDDs [14].

Nowadays, the research efforts are focused on the development of small-molecule-based compounds known as “multi-target-directed ligands” (MTDLs) [15,16]. Such drug molecules are able to modulate more than one pharmacologically relevant central nervous system (CNS) targets, while having low affinity against other cellular proteins. The MTDLs should also have suitable physicochemical and toxicological properties, including reduced risk of side effects [5,17]. Moreover, MTDLs potentially may have synergistic or additive effects having simultaneously a single pharmacokinetic and pharmacodynamics profile [5].

The in silico approaches have demonstrated their potential as a suitable tool to search for MTDLs as promising drug candidates with cost and time benefits. A study by Jaiteh et al. [18] shows the process of identifying dual-target ligands of the adenosine A_2A_ receptor (A_2A_R) and MAO-B for drug development against Parkinson’s disease. An initial set of 5.4 million compounds is docked separately in both receptor structures and then scored using DOCK3.6. The sum of the ranking from the two screenings is then used to narrow down the sample to the 500 lowest scoring compounds and after further screening—to 14 active for either MAO-B or A_2A_R. Among the 14 experimentally confirmed ligands, only four displayed activities in both receptors. Analysis of the docking results revealed structural similarities between the top-ranked ligands and the reference dual-acting A_2A_R antagonist and MAO-B inhibitor (*E*)-8-(3-Chlorostyryl)caffeine (known also as CSC) [18,19,20]. Remarkable for the research is also that one of the compounds while very potent MAO-B inhibitor (IC_50_ = 100 nM) happens to be one of the highest affinity A_2A_R ligands (*K_i_* = 19 nM).

In other research with a main focus on A_2A_R and MAO-B, Perez-Castillo et al. [21] propose a different approach. Their methodology includes docking of a set of molecules (including known dual-target ligands and decoys) to both receptors using six different scoring functions and subsequent rescoring of the docking poses. Then, depending on the combination of scoring functions, a fused rank is produced. In this way, a higher value of enrichment of known ligands is obtained compared to individual single function scoring. According to this study, a 22-fold increase of the enrichment factor of the top 1% of the final database ranking is observed using the fused scoring methodology.

A combined in silico approach is applied by Ganai et al. (2017) in order to identify selective HDAC2 inhibitors lacking HDAC1 inhibiting activity [22]. Based on their role in regulation the homeostasis of intracellular histone acetylation, HDAC1 is found to act as neuroprotective agent by enhancing the synaptic plasticity and neuronal survival [23,24], while HDAC2 is shown to downregulate the synaptic plasticity and memory formation [10,12]. Thus, Ganai et al. [22] dock structurally different groups of HDAC-inhibitors (hydroxymates, cyclic tetrapeptides and short-chained fatty acids) in HDAC1 and HDAC2. The results show that hydroxymates and cyclic tetrapeptides demonstrate different affinities towards HDAC1 and HDAC2. The molecular dynamics (MD) simulation shows that the presence of positive ionizable group plays a significant role in the selective inhibition of HDAC1 while its absence is beneficial for the HDAC2 inhibition.

Research by Khalid et al. focuses on virtual screening (VS) of biaryl-scaffold-containing compounds for possible anti-aggregatory and neuroprotective effect in AD [25]. The screening is carried out against common targets for AD—AChE, β-secretase (BACE1), monoamine oxidases (MAOs), and *N*-Methyl-D-aspartate (NMDA) receptors. Based on the performed docking it is concluded that very few ligands show strong affinity towards all targets, however several have notable interactions with at least two targets. The results suggest that biaryl scaffold, and more precisely biaryl sulphonamides, may be potential candidates for multi-target drug therapy in patients with AD.

Despite the recent efforts in searching multi-target anti-NDD drugs, the niche of the in silico studies remains largely unexplored yet in relation to the targets involved, the screened small molecules, and the combination of the methods applied. In this study we focused on a panel of three proteins involved in NDD pathology which combination hasn’t been explored so far, including MAO-B, AChE and HDAC2 (Figure 1).

We applied a variety of in silico structure- and ligand-based methods complemented by MD simulations to reliably evaluate the binding of to these proteins and therefore the pharmacological potential of a set of more than 650,000 small molecules, purposely constructed as a library of unique drug-like compounds. The subsequent analysis of the pharmacokinetic/safety profile of the top scored hits followed by MD simulations of their behavior in the active sites of the selected proteins revealed two drug-like compounds as potential multi-target drug candidates. Our results accomplish the first step in the selection of multi-target acting hits for subsequent drug development against NDDs.

## 2. Results and Discussion

The computational workflow of the study is presented in Figure 2.

### 2.1. Structure-Based Virtual Screening to Identify Potential Multi-Target Ligands

The advantages of the applications of VS approaches to identify promising hit compounds have been demonstrated in a number of studies [26,27,28]. For establishing a reliable in silico VS protocol in this work we first analyzed the X-ray complexes of the three enzymes and re-docked the reference ligands to set the score thresholds for selection of the drug-like compounds with appropriate docking energies (better than the re-docked reference ligands scores). The X-ray complexes’ binding sites and the specific ligand-protein interactions are presented in Figure 3. Specific interactions have been recorded for each ligand, outlining important amino acid residues in the active sites of the targeted proteins. This information was considered in the further analysis of the reference ligands and the hit compounds.

A cross-docking of the co-crystallized ligands in the active sites of each of the other two proteins was performed to elucidate the compatibility of the binding sites, i.e., the possibility of the ligands to be posed with a good docking score in the binding sites of all three proteins. The protocols selected for this cross-docking evaluation were those selected for initial filtering—MOE rigid docking and MOE flexible docking. The results are summarized in Table 1.

With the MOE rigid docking, we observed the HDAC2 active site compatibility to the ligands of MAO-B and AChE (cross-docking score ranges of safinamide and donepezil overlapped with the re-docking score range of the benzamide derivative). However, with MOE flexible docking, when the induced fit refinement was performed, the overlap of cross-docking and re-docking score ranges was observed for all three co-crystallized ligands in all three active sites. Inspection of the protein-ligand interactions in the cross-docking poses indicated reasonable orientations and conformations of the co-crystallized ligands. Specific interactions were observed in all studied complexes also including residues involved in interactions with the own ligand of the protein (Figure 3): Gln206 in MAO-B; Trp86 and Phe295 in AChE; Gly154 and Asp181 in HDAC2. Specific interactions with Zn^2+^ were recorded for safinamide and donepezil in the HDAC2 binding site. These observations confirmed the compatibility of the binding modes of the reference ligands in the MAO-B, AChE and HDAC2 and thus outlined a possibility to find triple-active compounds among those in the selected library.

After the analysis of the co-crystallized ligands, VS of the input CCG lead-like conformer database was performed in a number of steps to retrieve hits with a potential to interact in the active sites of the three investigated enzymes (outlined in Figure 3). All 653,214 compounds were docked by MOE rigid docking in the active site of each of the three enzymes. From them, 11,085 compounds were selected based on their docking scores that exceed those of the re-docked reference ligands for all the proteins. The subsequent application of the MOE flexible docking protocol on those 11085 compounds resulted in 1011 structures, which passed the triple-activity criteria according to the scores reported in Table 1 for this protocol.

Further, these 1011 structures were subjected to docking in SeeSAR and a putative affinity range (*K_i_*
_HYDE_ range) was calculated for each ligand applying HYDE post-docking procedure. This procedure resulted in elimination of 566 structures which were not docked successfully or *K_i_*
_HYDE_ range was not calculated for any of the three target proteins (Table 2). The remaining 445 compounds were further filtered in KNIME by applying sub-structural filters in order to remove chemical compounds known to show false-positive results by reacting non-specifically with numerous biological targets, and, therefore, with a high probability to be non-active against the target of interest [29]. Such frequent hitters or false-positive inhibitors are known as pan assay interference compounds (PAINS) or colloidal aggregators [30,31,32,33]. In addition, compounds containing a Schiff base substructure known to undergo a rapid degradation (hydrolysis) were also filtered from the set of compounds [34]. The above sub-structural filters were used to identify and remove from further screening sets the compounds that may interfere with biological assays due to their reactivity, interference with assay conditions, false activities (e.g., oxidizers, detergents), chemical instability (e.g., Schiff bases, latent aldehydes), and others [35].

The MTDL effect of the compounds was considered according to the predicted binding affinities as approximated by their HYDE scores. The dataset of 377 compounds after the PAINS filtering was analyzed based on the predicted affinity of the compounds to each individual protein. Figure 4 illustrates the distribution (in %) of these compounds according to the calculated affinity values for each of the three enzymes. The affinity threshold for each individual enzyme was set to be less than 5.0 µM thus narrowing the selection to 41% for AChE, 25% for HDAC 2 and 43% for MAO-B. The threshold value was chosen based on the understanding that for a hit molecule, the affinity is expected to be in the micromolar range [36]. The final selection was based on the multiplication product of the all compound’s three affinity values less than 100 × 10^6^.

The application of these affinity constraints resulted in 16 hits, which are shown in Table 3.

In general, all 16 hit compounds may represent good candidates for further in vitro screening. However, to narrow the range of experimental testing, the selection of hits was further refined using pharmacophore-based screening. For this purpose consensus pharmacophore models were built for each protein. A complex-based pharmacophore modeling approach was chosen, meaning that the best docking poses for each of the 16 compounds were extracted from each binding site and further used to generate a consensus pharmacophore model for each protein. The consensus pharmacophore of the 16 top scored hits showed three or more pharmacophore features for each protein with the aromatic/hydrophobic one present in all three of them (Figure 5).

### 2.2. BBB Penetration and Safety Profile Prediction of the Identified Hits

The database explored in this study consists of drug-like molecules, thus with generally suitable pharmacokinetic properties according to physicochemical parameters’ based rules such as Lipinski’s rule of five for oral compounds [37]. Furthermore, considering the fact that drugs used for NDD treatment have to be CNS-active, we subjected the four hits to an in silico estimation of their BBB penetration potential using the SwissADME web-based tool and the ACD/Percepta software platform (the results summarized in Table 4). Taking into account that the models differ in the coverage of the parameters/mechanisms predicted, a consensus final prediction was assigned to each compound in order to keep the different information provided by both models—the compound was classified as a “BBB penetrant” if one of the models classified it as such. For the purpose of the consensus classification the weak penetrants (in terms of ACD/Percepta predictions) were considered as “non-penetrants”. Based on the predictions (Table 4) the compound Asinex BAS 07211091 was considered as “non-penetrant” and excluded from further analyses.

Further, to assess the safety profile of the selected hits, Derek Nexus expert system was employed searching for specific structural alerts (toxicophores) associated with specific toxicities. In particular, liver and cardiac toxicities were predicted which are often identified as the main reason for toxicity-driven non-clinical safety-related drug candidate attrition [38]. Derek Nexus results provided predictions with “plausible” level of likelihood for the compound Chem T&I AMCLME-10390 to exhibit hepatotoxicity and cardiotoxicity in mammals (Table 4). Nevertheless, this compound was included in the subsequent MD simulations to investigate its interactions with the three enzymes. 

### 2.3. Molecular Dynamics Simulations of the Selected Hits

The retained three hits (Table 4) were subjected to MD simulations in order to investigate the stability of binding to the investigated enzymes. The calculated RMSDs of the displacements of the protein-ligand complexes over the simulation time are presented in Figure 6. They are indicators for the stability of the complexes. 

Although the position of donepezil in the complex with AChE had some large fluctuations around 50 and 90 ns of the simulation, the complex reached equilibrium after 175 ns with RMSD fluctuating within around 0.5 Å for the protein and within around 1 Å for the ligand. This was also valid for the complexes with the selected three hit compounds, after 175 ns stable complexes were obtained with the RMSD fluctuating within 0.5 Å for the proteins, and within 1 Å for the ligands. 

The MD results showed a stable complex of HDAC2 with the benzamide derivative—the RMSD for the protein fluctuated within less than 0.5 Å and the RMSD for the ligand fluctuated within around 1 Å after 150 ns of the simulations. The complex of HDAC2 with Specs AH-487/42478269 was less stable—the protein RMSD fluctuated within less than 0.5 Å, but the ligand position varied with RMSD within a range of around 3 Å at the end of the simulation. The complex of HDAC2 with Comgenex CGX-3274395 reached equilibrium after 180 ns with RMSD of the protein and ligand fluctuating within 0.5 Å and 1.0 Å, respectively. The complex of HDAC2 with Chem T&I AMCLME-10390 had strong fluctuations between 50 and 140 ns, at the end of the simulation it was stable with protein RMSD varying less than 0.5 Å and ligand RMSD varying within around 1.0 Å. 

The complexes with MAO-B reached equilibrium, with protein RMSD fluctuating within less than 0.5 Å and ligand RMSD within 1 Å or less after 160 ns. Exception was the complex of MAO-B with Chem T&I AMCLME-10390 where the ligand position reached equilibrium, but the protein position did not stabilize around a fixed RMSD value.

The protein-ligand interactions (PLIs) are presented in Figure 7, including 2D diagrams of the interaction points and histograms with the interacting amino acids. In the histograms the PLIs are categorized into four types—hydrogen bonds, hydrophobic, ionic, and water bridges. The histograms height represents the proportion of the simulation time (normalized to 1) during which the specific interaction was present (i.e., a value of 0.7 means that the interaction took place during 70% of the simulation time). Values over 1.0 were obtained in cases where the protein residue makes multiple contacts of the same type with the ligand. The 2D diagrams present specific interaction subtypes. Hydrophobic contacts are divided in Pi-cation, Pi-Pi, and other non-specific interactions. Water bridges are hydrogen-bonded PLIs mediated by a water molecule. In the 2D diagrams interactions that occurred during at least 20% of the simulation time, are shown.

For the complex of AChE with donepezil Trp86, Trp286, Phe295, Tyr337, Phe338, Tyr341, and His447 were shown to participate in the PLIs. Trp86, Trp286, and Phe295 were also outlined in the analysis of the PLIs of the co-crystalized reference ligands (Figure 3). The interactions with the three selected hits involved more protein residues, including the above residues, and other residues with longer interaction times, i.e., Glu202 for Specs AH-487/42478269, Tyr133 for Comgenex CGX-3274395, Asp74 and Tyr124 for Chem T&I AMCLME-10390. 

For the interaction of benzamide derivative with HDAC2 residues Asp104, Leu144, His145, His146, Gly154, Phe155, Asp181, His183, Phe210, Asp269, Gly307 and Tyr308 were outlined. Metal coordination interactions with Zn^2+^ involving Asp181, His183 and Asp269 were maintained during 100% of the simulation time. In the docking analysis Gly154, Asp181, and Tyr308 were shown to interact with the ligand (Figure 3). Some of the above residues took part in the interactions of Comgenex CGX-3274395 with HDAC2—Leu144, Gly154, Phe155, Asp181, His183, Phe210, Asp269, and Tyr308, including also interactions with Arg39 and Gly305. This complex showed to be stable by the RMSD values. Some of the above residues participated in the interactions of the complex of HDAC2 with Specs AH-487/42478269—Asp104, His146, Gly154, Phe155, Asp181, His183, Phe210, Asp269, including also Gly143. However, apart from the metal coordination with Zn^2+^ and the interaction with Phe210 (hydrophobic and H-bonding), the other interactions were present in less than 50% of the simulation time, which may explain the lower stability of the complex. For the complex of HDAC2 with Chem T&I AMCLME-10390 shown by the RMSD values also to be less stable, no PLIs were taking place during the whole period of the simulation—the interactions with the residues Asn100, His146, Glu151, Ala152, Phe155, His183, Phe210 and Tyr308 were present during or less than 50% of the simulation time, the interactions with Asp104 (hydrophobic, ionic, water bridge) were present around 70% of the simulation time. No metal coordination with Zn^2+^ was observed for this complex.

The residues interacting during almost 100% of the simulation time for MAO-B with safinamide were Leu171 and Tyr326. Leu171 and Gln206 were outlined above (Figure 3). Gln206 was also shown to interact by the MD simulations (through H-bonding and water bridge), but only in around 20% of the simulation time. Ile198, Ile199, Lys296, Ile316, Phe343, Tyr398 and Tyr435 also took part in the interactions in around 20% of the simulation time. Interactions with most of these residues were present in the other MAO-B complexes. For the complex with Specs AH-487/42478269 the interactions were with Leu167, Leu171, Gln206, Tyr326, Tyr398 and Tyr435. The complexes with Comgenex CGX-3274395 and Chem T&I AMCLME-10390 involved the same interacting residues as with safinamide except that Lys296 was excluded and Tyr188 was included in the interactions.

According to the above results, in general the enzyme-ligands complexes of the selected hits included similar interactions as those involved in the complexes with the crystallographic ligands. Some complexes involved more interaction points than the crystallographic ligands suggesting stronger binding compared to the reference ligands.

The binding energies of the enzyme ligand complexes were calculated from the most abundant cluster of the simulation trajectory poses. The calculated binding energies are presented in Table 5. The energies for the selected compounds are lower than the energies calculated for the crystallographic ligands confirming possible stable binding of these compounds to the investigated enzymes.

In general, the results from the MD simulations showed that the selected compounds may form stable complexes with the investigated enzymes involving interactions which resemble those in the crystallographic complexes. Exceptions are the complexes of Specs AH-487/42478269 and Chem T&I AMCLME-10390 with HDAC2, which may be less stable. In addition, the complex of Chem T&I AMCLME-10390 with MAO-B did not reach equilibrium at the simulation. 

Based on the above computational analyses, two of the final multi-target hits, Specs AH-487/42478269 and Comgenex CGX-3274395, were outlined as the most suitable lead structures for NDDs drug development.

## 3. Materials and Methods

### 3.1. Protein-Ligand Complexes

For the purposes of the investigation all crystallographic structures of MAO-B, AChE and HDAC2 available in the Protein Data Bank (PDB) [39] were analyzed. The final selection of protein-ligand complexes was carried out based on the X-ray structures’ resolution and availability of experimental data on the inhibitory effects of the co-crystallized ligands to be used as reference structures in the subsequent computational studies. The following complexes were selected as reference structures for VS: 4EY7 (AChE with donepezil; resolution 2.35 Å; binding affinity annotations in PDB: *K_i_*-min: 2.9, max: 38 nM from 9 assays) [40], 4LY1 (HDAC2 with a benzamide derivative; resolution 1.57 Å; binding affinity annotations in PDB: *K_i_*-min: 0.2, max: 1.5 nM from 2 assays) [41], and 2V5Z (MAO-B with safinamide; resolution 1.57 Å; binding affinity annotations in PDB: *K_i_* = 450 nM from one assay) [42]. For the purposes of the subsequent VS the B chains were selected from 4EY7 (for AChE) and 2V5Z (for MAO-B), which are homodimers, and the C chain from 4LY1 for HDAC2 that is homotrimer. The selection of these chains was based on the highest estimated ligand binding affinity to the particular chain using the post-scoring function HYDE (HYdrogen DEsolvation) in SeeSAR in order to predict the so called *K_i_*
_HYDE_ ranges (SeeSAR v. 10; BioSolveIT GmbH, Sankt Augustin, Germany, https://www.biosolveit.de/SeeSAR/, accessed on 22 August 2022) [43]. The cofactors flavin (in 2V5Z) and Zinc (in 4LY1), were kept in the structures for further computations [41,42].

### 3.2. Compounds Library

The initial dataset consisted of 653,214 unique small drug-like compounds taken from the Lead-like Conformer Database compiled by Chemical Computing Group (CCG) and provided with Molecular Operating Environment (MOE) platform (CCG Inc., Montreal, QC, Canada, https://www.chemcomp.com/, accessed on 22 August 2022). The structures in the database have been selected from 44 public catalogs of chemical suppliers, curated, protonated at physiological pH, and filtered for drug-likeness.

### 3.3. Docking and Virtual Screening

A multi-step computational workflow using different software and protocols for docking/VS was employed:

(i) Docking in MOE software v. 2020.09, using triangle matcher placement, London dG scoring function [44], no water molecules considered. Two subsequent runs were performed: with no refinement of the ligand (“MOE rigid-receptor docking”) and with induced fit refinement (flexible side chains—optimization of the ligand and the active site amino acids side chains) with GBVI/WSA dG scoring function [45] (“MOE flexible-receptor docking”).

(ii) Docking in SeeSAR software, using FlexX 4 placement [46], Böhm’s scoring function [47], subsequent induced fit refinement (flexible side chains) with HYDE scoring function [48]; water molecules in the active site were considered (“SeeSAR flexible-receptor docking”). Based on the HYDE score the affinity range of the docked ligands was estimated (*K_i_*
_HYDE_ range). The predicted *K_i_* is presented as a geometric mean of the lower and upper boundaries of the *K_i_*
_HYDE_ range.

(iii) Filtering-out the compounds with undesirable features (unstable compounds and those reacting non-specifically with numerous biological targets) in the KNIME analytics platform (https://www.knime.com/, accessed on 22 August 2022).

(iv) Pharmacophore analysis: consensus pharmacophores were generated in MOE from the top-scored ligands posed in the binding pocket of each of the three proteins. The applied settings were: neighborhood distance tolerance up to 1.45 Å, threshold 50%.

Each ligand was docked in the binding pocket of each of the three enzymes. Ten poses were generated for each ligand but only the best one was retained.

### 3.4. Selection of the Potential Multi-Target Ligands

The compounds exhibiting docking scores better than the best re-docking score of each of the three co-crystallized ligands in the receptor site of each enzyme were selected for the next VS step. Final ranking of the compounds was performed according to the multiplication product of the predicted *K_i_* values. The use of multiplication products, instead of sums, allows avoiding possible distortion due to differences in scaling of the docking scores in different proteins.

### 3.5. Blood-Brain Barrier Penetration Prediction

The hits, retrieved by the VS, were evaluated for their ability to cross blood-brain barrier (BBB). For this purpose, two in silico tools were employed:

(i) SwissADME web tool (http://www.swissadme.ch/, accessed on 22 August 2022). It predicts passive BBB-permeation based on a classification model including two physicochemical parameters—topological polar surface area (tPSA) and lipophilicity expressed as logP. According to the model compounds with moderate polarity (tPSA < 79 Å^2^) and lipophilicity (0.4 < logP < 6.0) possess a high probability to cross the BBB by passive diffusion and to access the CNS [49].

(ii) ACD/Percepta software (ACD/Labs Release 2021.2.2, Advanced Chemistry Development, Inc., Toronto, ON, Canada, https://www.acdlabs.com, accessed on 22 August 2022). This tool classifies the compounds as “penetrant”, “weak penetrant” and “non-penetrant” based on an equation, that is a combination of predicted quantitative characteristics of transport across BBB, namely brain/plasma equilibration rate and steady-state brain/plasma distribution ratio.

### 3.6. Safety Profile Elucidation

In silico prediction of the potential toxic effects of the retrieved hits was performed using Derek Nexus v.6.2.0 expert system (Lhasa Limited, Leeds, UK, https://www.lhasalimited.org, accessed on 22 August 2022). Derek Nexus generates a prediction by comparing the structural features of the compound with a toxicophore (structural alert) encoded as structural pattern(s) in its knowledge base. The final predictions are derived from a reasoning scheme which takes into account other relevant factors, for example physicochemical properties, as well as the presence of a toxicophore in the query structure [50]. The predictions in Derek Nexus are provided with the following levels of likelihood, from highest to lowest one: “certain”, “probable”, “plausible”, “equivocal”, “doubted”, “improbable”, and “impossible” [51]. In this study, the level of likelihood “plausible” was selected as a threshold, meaning “the weight of evidence supports the proposition”. The predictions were restricted to mammal species.

### 3.7. Molecular Dynamics Simulations

Schrödinger molecular modeling software (v. 2022.2, Schrödinger, Inc., New York, NY, USA https://www.schrodinger.com/, accessed on 30 June 2022) was used for MD simulations. Before the simulations “Protein Preparation” feature in Schrödinger with default settings was applied, including “Optimize H-bond Assignments” step. LigPrep feature with OPLS4 force field was used for ligand preparation. MD was done with Desmond module of Schrödinger [52]. The systems were prepared with “System Setup” feature. Solvent was added by applying “Water TIP3P” model in orthorhombic simulation box with minimum distance between the protein surface and the solvent surface of 10 Å. The systems were neutralized by adding Na^+^ or Cl^−^ as counter ions. A 0.15 M NaCl was used to obtain isosmotic environment. During the simulations Nose-Hoover thermostat at temperature 300 K and atmospheric pressure (1.013 bars), with the default setting for relaxation before simulation, were applied. The simulations were performed for a total of 200 ns, and 5000 frames were saved. The trajectories were examined with “Simulation Interaction Diagram” module of Desmond.

MM-GBSA tool in Prime module of Schrödinger was used to calculate the protein-ligand binding energies. For this purpose, the trajectory frames were clustered with the “Desmond Trajectory Clustering” module. All 5000 frames were used to generate (MSD matrix based on the protein backbone and the ligand positions (including Zn^2+^ for HDAC2). The representative of the most abundant cluster for each protein-ligand complex was used to calculate the binding energies by MM-GBSA tool with OPLS4 force field. The “VSGB solvation model” was applied for the calculations (default setting, water solvent).

## 4. Conclusions

In this study, we applied a series of computational and ranking procedures to identify suitable multi-target drug candidates for NDDs symptomatic treatment. We focused on three important enzymes involved in the NDD pathology—AChE, MAO-B and HDAC-2, the combination of which as a multi-target goal has not been explored so far. We stepped on a database of more than 650,000 compounds pre-screened for their physicochemical drug-like properties, thus reducing the initial signal-to-noise ratio. The compounds with decent affinities for all studied targets were selected and passed through additional analyses to remove possible false-positive hits or known non-specifically binding compounds. This multi-step procedure yielded 16 drug-like compounds with predicted activity simultaneously towards AChE, MAO-B, and HDAC-2. Consensus pharmacophore models were developed for each protein and the structures that contained all pharmacophoric features were selected as potential multi-target hits. From them, one was predicted as BBB non-penetrant. The remaining three hits were subjected to MD simulations. The MD results for one of the hits suggested a lower potential for binding to two of the three enzymes. In addition, it was predicted as a potential hepatotoxic and cardiotoxic agent. The MD results for two of the ligands were promising thus yielding two final hits with possible multi-target action.

In summary, our results outline the potential of the in silico approaches as a rational basis for the development of multi-target compounds against NDDs.

## Figures and Tables

**Figure 1 ijms-23-13650-f001:**
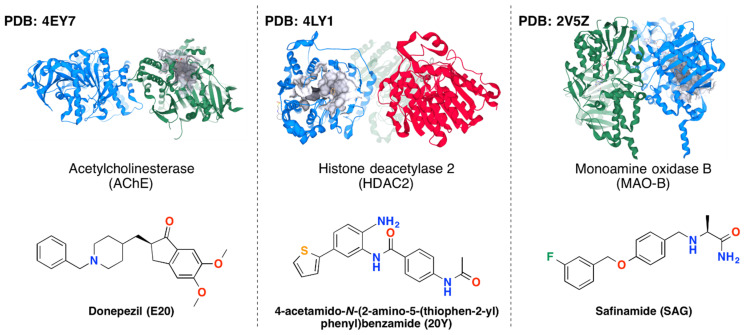
PDB crystallographic structures and reference ligands of the selected target protein complexes.

**Figure 2 ijms-23-13650-f002:**
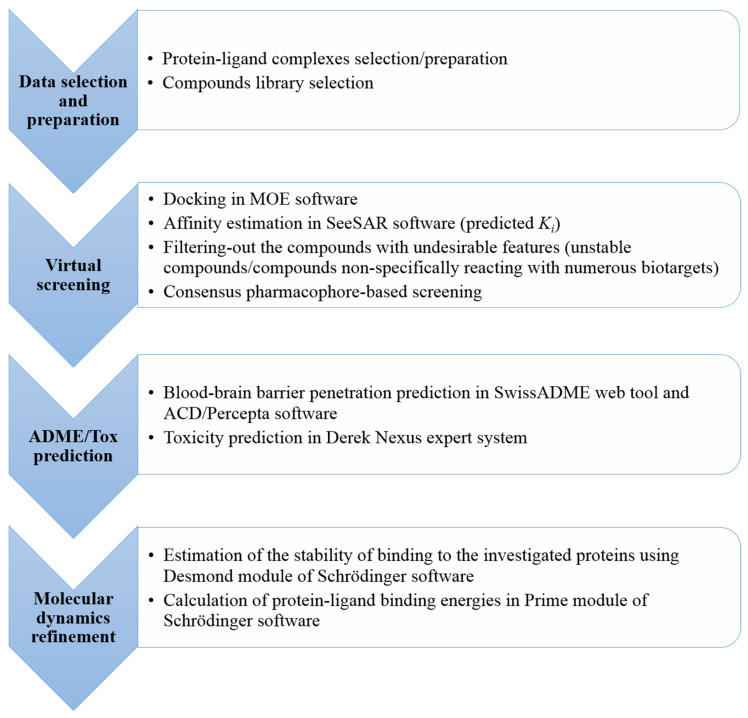
A flowchart representing the computational workflow with the main steps/approaches applied to identify the hit compounds toward AChE, HDAC2 and MAO-B enzymes.

**Figure 3 ijms-23-13650-f003:**
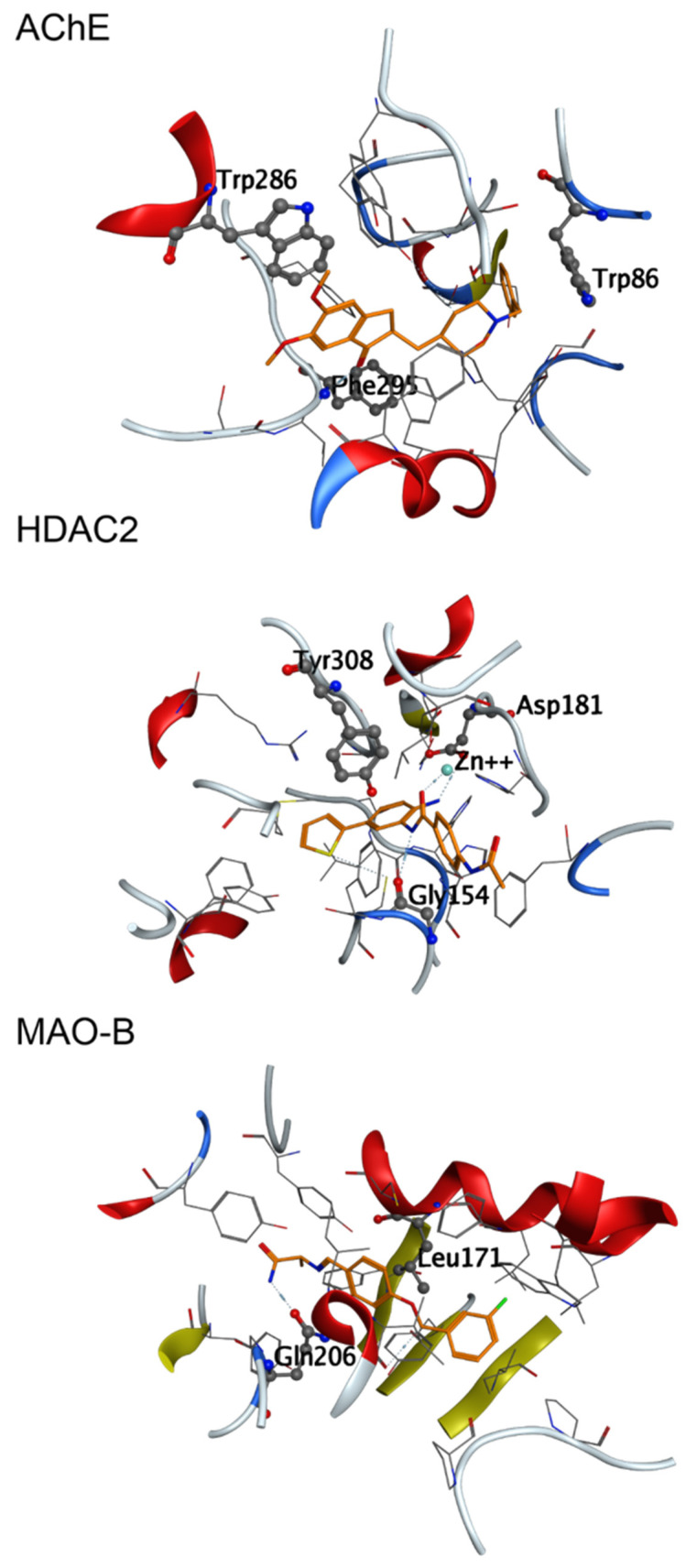
Active sites together with the bound reference ligands (in orange) of the selected proteins: the amino acids involved in specific interactions are presented in sticks & balls: AChE (Trp86, Trp286, Phe295); HDAC2 (Gly154, Asp181, Tyr308 and Zn^2+^ cofactor, represented as a sphere); MAO-B (Leu171, Gln206).

**Figure 4 ijms-23-13650-f004:**
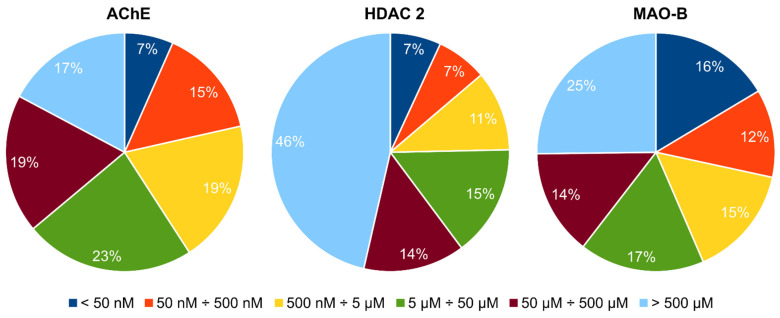
Distribution (in %) of the top 377 compounds based on the predicted *K_i_* values for each of the three proteins.

**Figure 5 ijms-23-13650-f005:**
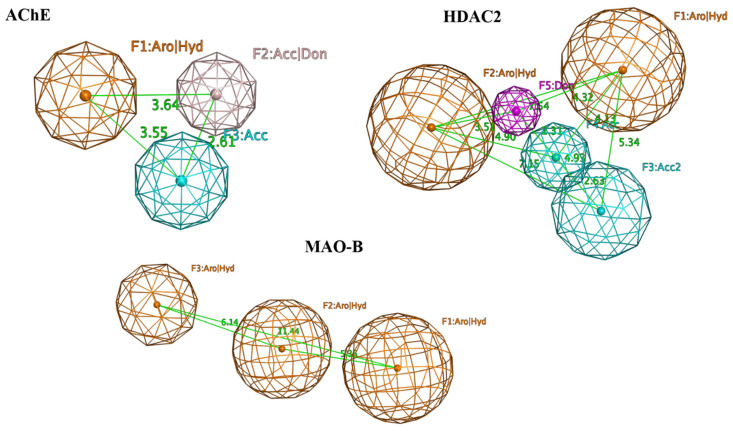
Consensus pharmacophore models of the top 16 hit ligands of AChE, HDAC2, and MAO-B. The following pharmacophoric features are outlined in the models: AChE—F1 (hydrophobic/aromatic feature), F2 (functional groups capable of performing hydrogen bonds (HB)), F3 (functional groups capable of performing HBs as HB acceptors); HDAC2—F1, F2 (hydrophobic/aromatic features), F3, F4 (functional groups capable of performing HBs as HB acceptors), F5 (functional groups capable of performing HBs as HB donors); MAO-B—F1, F2, F3 (hydrophobic/aromatic features).Individually for each protein, the pharmacophore features were present in 11 ligands for MAO-B, 8 for HDAC2 and 7 for AChE. Finally, four ligands contained every single pharmacophore feature for each protein (compounds: Specs AH-487/42478269 (1), Comgenex CGX-3274395 (8), Chem T&I AMCLME-10390 (10), and Asinex BAS 07211091 (16) in Table 3). These compounds were outlined as potential multi-target acting hits and subjected to further analyses as described below.

**Figure 6 ijms-23-13650-f006:**
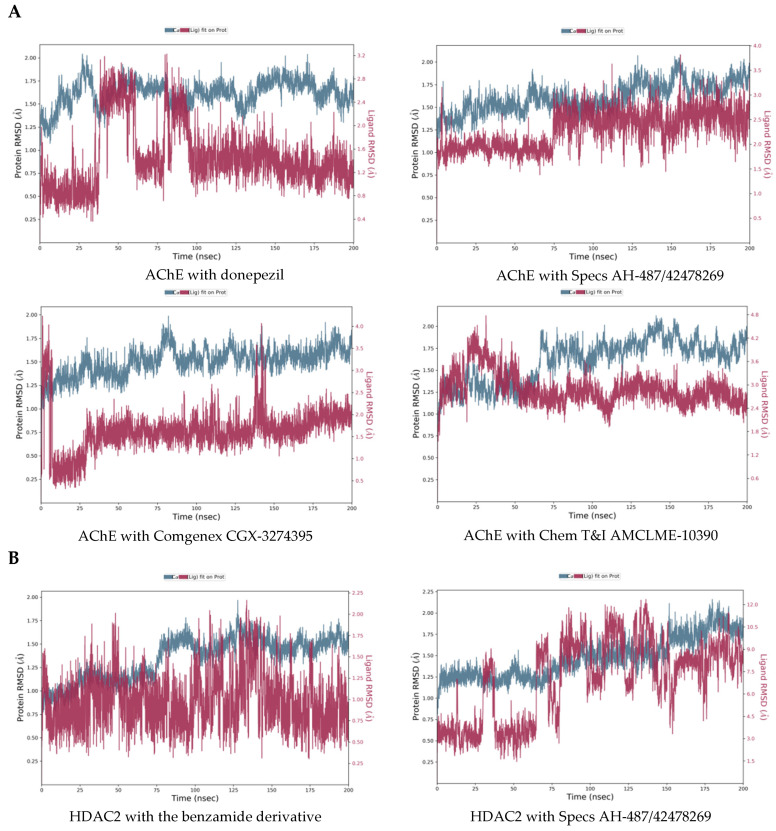
Calculated RMSDs (Å) of the displacements of the protein-ligand complexes over the MD simulation time: (**A**) AChE with reference ligand/hit ligands; (**B**) HDAC2 with reference ligand/hit ligands; (**C**) MAO-B with reference ligand/hit ligands.

**Figure 7 ijms-23-13650-f007:**
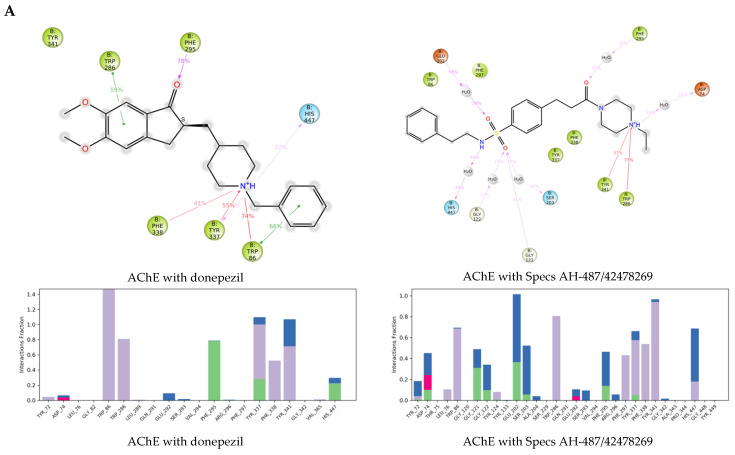
Protein-ligand interactions for the enzyme complexes with reference ligand/hit ligands obtained in the MD simulations: (**A**) AChE; (**B**) HDAC2; (**C**) MAO-B.

**Table 1 ijms-23-13650-t001:** Re-docking (in bold) and cross-docking scores of the co-crystallized ligands in the three enzymes using two docking protocols in MOE; the ranges span the scores of the best 10 poses. The lowest RMSD values (root mean square deviations; MOE rigid/flexible docking protocol) for the re-docked ligands (in bold) are the following: AChE—1.19 Å/0.93 Å; HDAC2—1.29 Å/0.66 Å; MAO-B—1.71 Å/0.78 Å.

Enzyme (PDB Code)	Docking Protocol	Docking Scores of the Reference Ligands, kcal/mol		
		Safinamide	Benzamide Derivative	Donepezil
**AChE (4EY7)**	MOE rigid	−13.05 ÷ −11.16	−12.13 ÷ −11.43	**−** **14.71 ÷ ** **−** **12.64**
	MOE flexible	−7.35 ÷ −5.83	−7.84 ÷ −6.62	**−** **8.52 ÷ ** **−** **6.29**
HDAC2 (4LYI)	MOE rigid	−11.72 ÷ −10.42	**−** **13.01 ÷ ** **−** **11.30**	−13.24 ÷ −10.92
	MOE flexible	−7.17 ÷ −5.62	**−** **7.68 ÷ ** **−** **6.74**	−8.51 ÷ −6.10
MAO-B (2V5Z)	MOE rigid	**−13.56 ÷ −12.45**	−7.84 ÷ −4.66	−9.12 ÷ −5.28
	MOE flexible	**−8.34 ÷ −7.69**	−9.08 ÷ −8.17	−9.96 ÷ −8.31

**Table 2 ijms-23-13650-t002:** Results of the multi-step VS with the number of triple-acting compounds remaining after each step.

Virtual Screening Steps	Number of Docked Compounds	Number of Compounds Passing the Criteria for Inclusion in the Next Step
MOE rigid docking	653,214	11,085
MOE flexible docking	11,085	1011
SeeSAR flexible docking	1011	445
PAINS filtering	445	377
Affinity constraints	377	16
Pharmacophore filtering	16	4

**Table 3 ijms-23-13650-t003:** Selected hits with their names, structures (incl. SMILES), affinity values (predicted *K_i_*, nM) and affinities’ multiplication product. Compounds are ranked according to the *K_i_* multiplication product.

No.	Name/Structure	*K_i_* Predicted (nM)			Multi-Plication Product
		AChE	HDAC2	MAO-B	
1	Specs AH-487/42478269 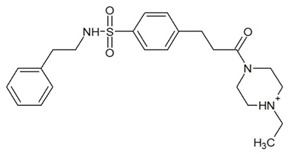 S(=O)(=O)(NCCc1ccccc1)c2ccc(cc2)CCC(=O)N3CC[NH+](CC3)CC	131.0	5.971	0.6379	497.4
2	Tripos 1503-03309 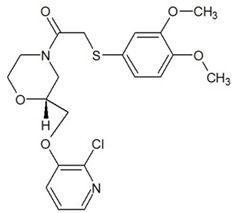 Clc1ncccc1OC[C@@H]2OCCN(C(=O)CSc3cc(OC)c(OC)cc3)C2	3.808	1615	1.784	10,970
3	Chembridge 7905648 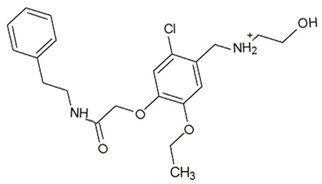 Clc1c(cc(OCC)c(OCC(=O)NCCc2ccccc2)c1)C[NH2+]CCO	4346	6.160	0.5084	13,610
4	Tripos 1526-25537 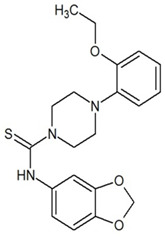 S=C(N1CCN(c2c(OCC)cccc2)CC1)Nc3cc4OCOc4cc3	17.04	868.2	0.9270	13,720
5	EMC Microcollections 010F0838 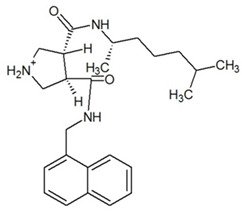 O=C(N[C@H](CCCC(C)C)C)[C@@H]1[C@@H](C(=O)NCc2c3c(ccc2)cccc3)C[NH2+]C1	337.9	3339	0.1196	135,000
6	Akos LT-1164 X 260 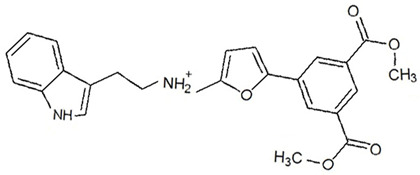 O=C(OC)c1cc(cc(c1)C=2OC(=CC2)C[NH2+]CCC=3c4c(NC3)cccc4)C(=O)OC	59.21	937.0	3.000	166,400
7	Chembridge 7928210 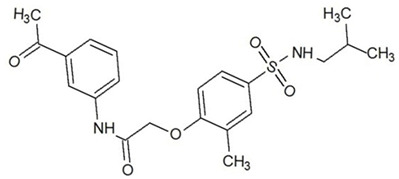 S(=O)(=O)(NCC(C)C)c1cc(c(OCC(=O)Nc2cc(ccc2)C(=O)C)cc1)C	109.6	11.56	309.0	391,700
8	Comgenex CGX-3274395 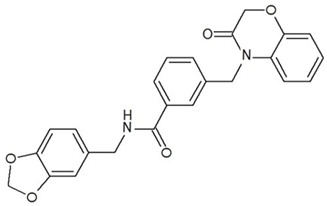 O=C(NCc1cc2OCOc2cc1)c3cc(ccc3)CN4c5c(OCC4=O)cccc5	14.15	13.76	3658	712,200
9	Chem T&I SHCLME-048161 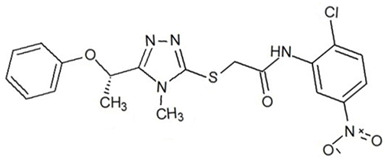 Clc1c(NC(=O)CSC2=NN=C(N2C)[C@@H](Oc3ccccc3)C)cc([N^+^]([O-])=O)cc1	34.19	72.17	439.6	1,085,000
10	Chem T&I AMCLME-10390 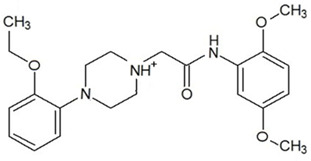 O=C(Nc1c(OC)ccc(OC)c1)C[NH+]2CCN(c3c(OCC)cccc3)CC2	335.9	930.0	5.961	1,862,000
11	Chemdiv 4378-0361 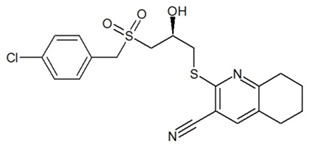 Clc1ccc(cc1)CS(=O)(=O)C[C@@H](O)CSc2nc3c(cc2C#N)CCCC3	88.81	517.3	93.30	4,287,000
12	Tripos 1547-01361 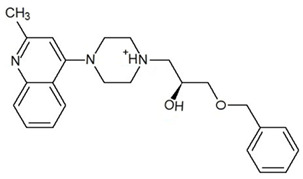 O(Cc1ccccc1)C[C@@H](O)C[NH+]2CCN(c3c4c(nc(c3)C)cccc4)CC2	4785	201.3	9.489	9,142,000
13	Chem T&I AMCLME-01759 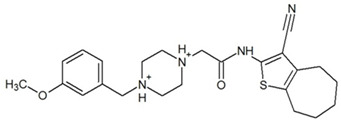 S1C(NC(=O)C[NH+]2CC[NH+](Cc3cc(OC)ccc3)CC2)=C(C#N)C4=C1CCCCC4	727.6	2412	5.263	9,238,000
14	Asinex ASN 04448308 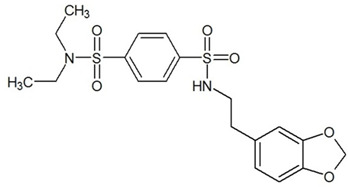 CCN(CC)S(=O)(=O)c1ccc(cc1)S(=O)(=O)NCCc1ccc2OCOc2c1	2203	323.0	52.53	37,370,000
15	Princeton Biomolecular OSSK_456453 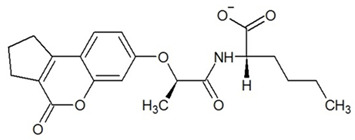 [O-]C(=O)C(CCCC)NC(=O)C(C)Oc1ccc2c(c1)OC(=O)C=1CCCC2=1	65.99	1545	420.6	42,880,000
16	Asinex BAS 07211091 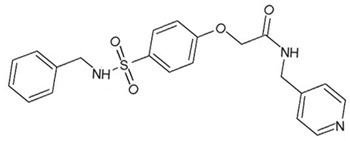 O=S(=O)(NCc1ccccc1)c1ccc(cc1)OCC(=O)NCc1ccncc1	4794	6.599	2053	64,960,000

**Table 4 ijms-23-13650-t004:** BBB penetration and toxicity predictions for the hits identified in the virtual screening.

Structure/Name	BBB Prediction		BBB Prediction (Consensus)	Derek Nexus Toxicity Prediction
	SwissADME	ACD/Percepta		
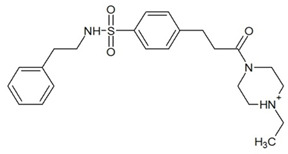 1. Specs AH-487/42478269	non-penetrant	penetrant	penetrant	No
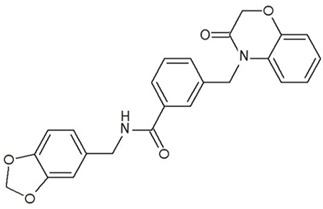 2. Comgenex CGX-3274395	penetrant	weak penetrant	penetrant	No
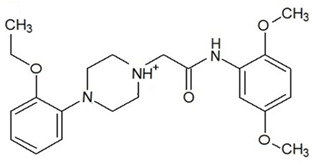 3. Chem T&I AMCLME-10390	non-penetrant	penetrant	penetrant	hepatotoxicity cardiotoxicity ^a^
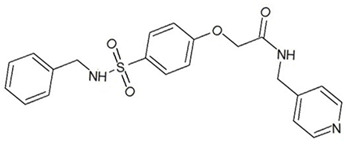 4. Asinex BAS 07211091	non-penetrant	weak penetrant	non-penetrant	No

^a^ Cardiotoxicity endpoint: HERG channel inhibition in vitro.

**Table 5 ijms-23-13650-t005:** Binding energies of the protein-ligand complexes from the MD simulations (kcal/mol).

Ligand	AChE	HDAC2	MAO-B
Crystallographic ligand	−71.49	−21.41	−45.29
Specs AH-487/42478269	−76.01	−30.16	−77.37
Comgenex CGX-3274395	−72.83	−45.41	−83.35
Chem T&I AMCLME-10390	−82.14	−48.94	−74.57

## Data Availability

The raw data are available on request from the authors.

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
