# Peer review of "In Silico Identification of Multi-Target Ligands as Promising Hit Compounds for Neurodegenerative Diseases Drug Development"

_ijms, 2022, doi:10.3390/ijms232113650_

Round 1

Reviewer 1 Report

General comment

The idea of searching “multi-target-directed ligands” (MTDLs) is interesting, but their application in neurodegenerative disorders (NDDs) may not be as straight forward as the authors described. The treatments for adult neurodegenerative diseases like Parkinson’s disease and Alzheimer disease have been successful yet. Protein targets including acetylcholinesterase 19 (AChE), histone deacetylase 2 (HDAC2), and monoamine oxidase B (MAO-B) have been attempted, but none of them clearly alters the natural course of these diseases. Although a combinatory therapy may be helpful, but it should be more important to demonstrate the efficacy of any specific treatment. Therefore, this manuscript should be more a technical attempt rather than a drug development work. The title and content of this manuscript may need modifications.

Specific comments

Table 2. Is the “Number of docked compounds” for any of the three proteins or considered as MTDLs? Why Fig. 3 analyzes the 377 (these compounds look nonspecific or MTDL effect has been considered?), while afterward analyses are for the 16?  

Table 3. The Ki of each compounds for each of the three protein vary quite widely, so it is hard to consider them as MTDL. 

Figure 4. The legend is short and the figure is difficult to understand.

Author Response

Dear Editors,

We would like to thank the reviewers for their valuable remarks and comments that have been addressed in the revised version of our manuscript.

Below all points raised by the reviewers (in Italic) are answered and the corresponding additions/corrections in the manuscript are described in Track changes mode.

We hope that our revision will meet the expectations of the reviewers.

Looking forward to your decision concerning the revised version of our manuscript.

Kind regards,

Ivanka Tsakovska and Nikolay Tzvetkov

Reviewer 1

General comment

 The idea of searching “multi-target-directed ligands” (MTDLs) is interesting, but their application in neurodegenerative disorders (NDDs) may not be as straight forward as the authors described. The treatments for adult neurodegenerative diseases like Parkinson’s disease and Alzheimer disease have been successful yet. Protein targets including acetylcholinesterase 19 (AChE), histone deacetylase 2 (HDAC2), and monoamine oxidase B (MAO-B) have been attempted, but none of them clearly alters the natural course of these diseases. Although a combinatory therapy may be helpful, but it should be more important to demonstrate the efficacy of any specific treatment. Therefore, this manuscript should be more a technical attempt rather than a drug development work. The title and content of this manuscript may need modifications.

We thank the Reviewer for this important remark. We fully agree with the opinion that the search for MTDLs is not a straightforward task due to the complex nature of the NDDs. We also agree that manipulation of either of the selected proteins itself would not change the natural course of the neurodegenerative diseases (NDDs), e.g., Alzheimer´s or Parkinson´s disease. The choice of these three enzymes was motivated by the available experimental studies indicating the involvement of HDAC enzymes family in memory processes, as well as availability of already approved drugs, such as safinamide and donepezil, targeting MAO-B and AChE enzymes, respectively. Therefore, we were motivated by the reasoning that affecting simultaneously disease-modifying (HDAC2) and symptomatic treatment (AChE and MAO-B) targets could have presumably additive effect(s) and thus improve the NDDs treatment. In addition, such combination has not been investigated yet. To be more explicit in the background information about involvement of the selected proteins in the NDDs an additional text and references waere included in the Introduction.

In fact, the title as well as the content of our manuscript clearly outline that our study is focused on “in silico” identification of multi-target ligands related to NDDs and we applied a number of in silico methods and tools to narrow the range of the potential MTDLs from physicochemical, safety, and bioactivity point of view. In addition, we focus on the full possibilities these methods can provide in accordance with the topic of the special issue of IJMS considering at the same time the peculiarities of the NDDs.

We, however, agree with the Reviewer that the title is not fully appropriate and we modified it to become more specific and exact as follows:

In silico identification of multi-target ligands as promising hit compounds for neurodegenerative diseases drug development

We further improved the manuscript by giving more detailed examples of combined or mono therapy of the most prevalent neurodegenerative disease. All improvements are respectively indicated in the iItroduction part of the manuscript.

 Specific comments

  • Table 2. Is the “Number of docked compounds” for any of the three proteins or considered as MTDLs?

All 653214 compounds have been docked in each of the three enzymes and the passed 11085 compounds have been selected based on the docking scores that exceed those of the redocked ligands for all of the proteins

To make this explanation clearer the respective text in the Results and Discussion section has been modified as follows:

"After the analysis of the co-crystallized ligands, VS of the input CCG lead-like conformer database was performed in a number of steps to retrieve hits with a potential to interact in the active sites of the three investigated enzymes (outlined in Figure 2). All 653214 compounds were docked by MOE rigid docking in the active site of each of the three enzymes. From them 11085 compounds were selected based on their docking scores that exceed those of the redocked reference ligands for all the proteins. The subsequent application of the MOE flexible docking protocol on those11085 compounds resulted in 1011 structures, which passed the triple-activity criteria according to the scores reported in Table 1 for this protocol."

  • Why Fig. 3 analyzes the 377 (these compounds look nonspecific or MTDL effect has been considered?), while afterward analyses are for the 16?

The MTDL effect has been considered according to the binding affinities of the compounds as approximated by their docking scores. The group of the 16 compounds has been filtered out from the 377 compounds based on their predicted affinity for each individual protein (less than 5000 nM).

To make it clearer the corresponding text has been moved to appear before Table 2. Figure 3 (the new Figure 4) has been modified to reflect the threshold value and an explanation has been added in the text as follows:

"The MTDL effect of the compounds was considered according to the predicted binding affinities as approximated by their HYDE scores. The dataset of 377 compounds after the PAINS filtering was analyzed based on the predicted affinity of the compounds to each individual protein. Figure 4 illustrates the distribution (in %) of these compounds according to the calculated affinity values for each of the three enzymes. The affinity threshold for each individual enzyme was set to be less than 5000 nM thus narrowing the selection to 41% for AChE, 25% for HDAC 2 and 43% for MAO-B. The threshold value was chosen based on the understanding that for a hit molecule the affinity is expected to be in the micromolar range [50]. The final selection was based on the multiplication product of the all compound’s three affinity values less than 100×106."

"The application of these constraints resulted in 16 hits, which are shown in Table 3."

  • Table 3. The Ki of each compounds for each of the three protein vary quite widely, so it is hard to consider them as MTDL.

As explained above the affinity of each of the selected 16 hit compounds is below 5 µM that is considered a good value for a hit molecule which affinity is expected to be in the micromolar range. This was clarified in the text by adding an appropriate reference (DOI:10.1111/j.1476-5381.2010.01127.x).

  • Figure 4. The legend is short and the figure is difficult to understand.

The legend of the figure was detailed by explaining the information available on the figure (the new Figure 5).

Reviewer 2 Report

“The compounds with sufficient affinities for all studied targets were additionally filtered to remove known non-specifically binding or 21 unstable compounds.”

What is sufficient, replace this word

In silico should be italic

“The selected four hits underwent … blood-brain barrier penetration and safety evaluation, and molecular dynamics simulations …active compounds against NDDs.” Check this sentence

Several grammatical errors we noticed throughout the manuscript

The introduction part looks more like a discussion.  Need to be revised

Material and Method:

“For the purposes of the investigation all available in the Protein Data Bank (PDB) ….computational studies.”

Rewrite this sentence, not clear and too long sentence

Provide a flowchart of the complete study.

Was the redocked conformation of the ligan was similar to its crystal conformation? The analysis with RMSD difference needs to be provided.

The computational screening of the library was the starting point of this study and it should be discussed more in the discussion. Authors may  refer to several related articles:

DOI: 10.2174/1381612822666151125000550

DOI: 10.1007/s12272-015-0640-5

 DOI: 10.1093/bib/bbp023

The discussion part needs to be thoroughly revised. 

Author Response

Dear Editors,

We would like to thank the reviewers for their valuable remarks and comments that have been addressed in the revised version of our manuscript.

Below all points raised by the reviewers (in Italic) are answered and the corresponding additions/corrections in the manuscript are described in Track changes mode.

We hope that our revision will meet the expectations of the reviewers.

Looking forward to your decision concerning the revised version of our manuscript.

Kind regards,

Ivanka Tsakovska and Nikolay Tzvetkov

Reviewer 2

  • “The compounds with sufficient affinities for all studied targets were additionally filtered to remove known non-specifically binding or 21 unstable compounds.”

What is sufficient, replace this word

The word has been replaced and the value has been specified according to the reviewer's recommendation.

  • In silico should be italic

Although the italic font for "in silico" is commonly used as correctly specified by the Reviewer, the style of the Journal of Molecular Sciences requires non-italic font for these terms.

  • “The selected four hits underwent … blood-brain barrier penetration and safety evaluation, and molecular dynamics simulations …active compounds against NDDs.” Check this sentence

The sentence has been corrected according to the Reviewer's recommendation as follows:

"The selected four hits underwent subsequent refinement through in silico blood-brain barrier penetration estimation, safety evaluation, and molecular dynamics simulations resulting in two hit compounds that constitute a rational basis for further development of multi-target ac-tive compounds against NDDs."

  • Several grammatical errors we noticed throughout the manuscript

The manuscript has been carefully revised to correct the grammatical errors.

  • The introduction part looks more like a discussion.  Need to be revised.

The introduction part has been carefully revised to provide the necessary background information of the study.

  • Material and Method:

- “For the purposes of the investigation all available in the Protein Data Bank (PDB) ….computational studies.”. Rewrite this sentence, not clear and too long sentence

The sentence was modified according to the Reviewer's recommendation as follows:

"For the purposes of the investigation all crystallographic structures of MAO-B, AChE and HDAC2 available in the Protein Data Bank (PDB) [26] were analyzed. The final selection of protein-ligand complexes was carried out based on the X-ray structures' resolution and availability of experimental data on the inhibitory effects of the co-crystallized ligands to be used as reference structures in the subsequent computational studies."

- Provide a flowchart of the complete study.

A flowchart of the complete study has been provided according to the Reviewer's recommendation (see the new Figure 2).

- Was the redocked conformation of the ligan was similar to its crystal conformation? The analysis with RMSD difference needs to be provided.

To meet this requirement of Reviewer 2 the RMSD values have been provided in the text to Table 1 for the re-docked ligands as only possible. These are the lowest RMSD values for the redocked ligands following MOE rigid and MOE flexible protocol. As seen from the legend the values are below 2 Å and correspond to the more precise MOE flexible docking protocol.

  • The computational screening of the library was the starting point of this study and it should be discussed more in the discussion. Authors may  refer to several related articles:

DOI: 10.2174/1381612822666151125000550

DOI: 10.1007/s12272-015-0640-5

 DOI: 10.1093/bib/bbp023

The power of the VS approaches in identifying promising hit compounds has been discussed and appropriate references were added at the beginning of the Results and Discussion section according to the Reviewer's recommendation.

  • The discussion part needs to be thoroughly revised. 

The Results and Discussion section has been carefully revised following the Reviewers' recommendations.

Round 2

Reviewer 2 Report

Most of the raised comments are addressed.